# Urban Neighbourhood Environments, Cardiometabolic Health and Cognitive Function: A National Cross-Sectional Study of Middle-Aged and Older Adults in Australia

**DOI:** 10.3390/toxics10010023

**Published:** 2022-01-07

**Authors:** Ester Cerin, Anthony Barnett, Jonathan E. Shaw, Erika Martino, Luke D. Knibbs, Rachel Tham, Amanda J. Wheeler, Kaarin J. Anstey

**Affiliations:** 1Mary MacKillop Institute for Health Research, Australian Catholic University, Melbourne, VIC 3000, Australia; Anthony.Barnett@acu.edu.au (A.B.); Rachel.Tham@acu.edu.au (R.T.); Amanda.Wheeler@acu.edu.au (A.J.W.); 2School of Public Health, The University of Hong Kong, Hong Kong, China; 3Baker Heart and Diabetes Institute, Melbourne, VIC 3004, Australia; jonathan.shaw@baker.edu.au; 4Department of Community Medicine, UiT the Artic University of Norway, 9019 Tromsø, Norway; 5School of Public Health and Preventive Medicine, Monash University, Melbourne, VIC 3004, Australia; 6School of Life Sciences, La Trobe University, Melbourne, VIC 3086, Australia; 7School of Population and Global Health, University of Melbourne, Melbourne, VIC 3053, Australia; Erika.l.martino@unimelb.edu.au; 8Sydney School of Public Health, The University of Sydney, Camperdown, NSW 2006, Australia; luke.knibbs@sydney.edu.au; 9Menzies Institute for Medical Research, College of Health and Medicine, University of Tasmania, Hobart, TAS 7000, Australia; 10School of Psychology, University of New South Wales, Randwick, NSW 2052, Australia; k.anstey@unsw.edu.au; 11Neuroscience Research Australia (NeuRA), Sydney, NSW 2031, Australia

**Keywords:** walkability, greenspace, blue space, cardiometabolic health, cognitive function

## Abstract

Population ageing and urbanisation are global phenomena that call for an understanding of the impacts of features of the urban environment on older adults’ cognitive function. Because neighbourhood characteristics that can potentially have opposite effects on cognitive function are interdependent, they need to be considered in conjunction. Using data from an Australian national sample of 4141 adult urban dwellers, we examined the extent to which the associations of interrelated built and natural environment features and ambient air pollution with cognitive function are explained by cardiometabolic risk factors relevant to cognitive health. All examined environmental features were directly and/or indirectly related to cognitive function via other environmental features and/or cardiometabolic risk factors. Findings suggest that dense, interconnected urban environments with access to parks, blue spaces and low levels of air pollution may benefit cognitive health through cardiometabolic risk factors and other mechanisms not captured in this study. This study also highlights the need for a particularly fine-grained characterisation of the built environment in research on cognitive function, which would enable the differentiation of the positive effects of destination-rich neighbourhoods on cognition via participation in cognition-enhancing activities from the negative effects of air pollutants typically present in dense, destination-rich urban areas.

## 1. Introduction

Population ageing and urbanisation are global phenomena with major consequences for almost all sectors of society [1,2]. The majority of older adults live in cities [3], with migration from rural to urban areas expected to continue in this age group [2]. These trends call for an understanding of the impacts of urban environments on the health and well-being of older people.

One of the most burdensome non-communicable diseases in people aged 60 years or over is dementia [4]. Due to the current global increasing trend in population ageing, the number of dementia cases is expected to double every 20 years and reach over 135 million worldwide in 2050, with major societal implications for both high-income as well as low-middle-income countries [5]. An important cornerstone of a global strategy for long-term dementia risk reduction is cognitive health promotion throughout adulthood, which aims to maintain cognitive function in healthy individuals and minimise cognitive decline in those with cognitive impairment by targeting relevant modifiable risk factors [6]. These factors include several indicators of cardiometabolic health, such as obesity, elevated blood glucose and blood pressure [6] and dyslipidaemia [7].

Urban environments have been identified as important contributors to cardiometabolic health [8] through their influence on health-related lifestyle behaviours (e.g., physical activity and sedentary behaviours) [9,10] and through exposure to environmental stressors, such as air pollutants and noise [11,12,13]. More compact neighbourhoods with interconnected streets and access to amenities and parks were found to promote walking for transport [9] and leisure-time physical activity [14] in older populations and, by doing so, reduce sedentary behaviour [10,15]. A recent systematic review reported beneficial effects of walkable neighbourhoods (typified by higher levels of density, street connectivity and access to services) on changes in cardiometabolic health outcomes, including obesity, type 2 diabetes and hypertension [8]. There is, however, a dearth of studies on neighbourhood environmental characteristics potentially contributing to middle-aged and older adults’ cognitive function and the related role of cardiometabolic risk factors [16,17].

In general, studies on environmental correlates of cognitive function examined a limited range of neighbourhood attributes and often reported mixed findings [16], possibly because of the omission of important environmental confounders [17]. For example, while access to amenities is expected to benefit cognitive function by facilitating social engagement and promoting an active lifestyle and, hence, cardiometabolic health, it is also often associated with higher levels of traffic-related air pollution [18]. In this case, the omission of air pollution indicators from models of cognitive function may lead to the conclusion that destination accessibility is not an important factor [17]. This shortcoming can be addressed by including key sets of environmental attributes in models of cognitive function and considering their interrelationships, as proposed in an ecological model of urban environment effects on cognitive health [17,19], a simplified version of which, adapted to the present study, is presented in Figure 1.

The model in Figure 1 posits that characteristics of the urban built, natural and socio-economic environment and their by-products (air pollution), influence cognitive health indirectly via cardiometabolic health parameters that have been linked to dementia [6,7] and via other pathways (e.g., engagement in social and intellectual activities, attentional restauration) here depicted as ‘direct’ effects. Using this ecological framework, the present study examined the extent to which cardiometabolic health indicators explain associations of interrelated characteristics of the neighbourhood environment with cognitive function in mid-age and older Australians.

## 2. Materials and Methods

We used data from the Australian Diabetes, Obesity and Lifestyle (AusDiab) study, a three-wave population-based survey designed to examine the prevalence, incidence and determinants of diabetes in Australian adults aged 25 years and over [20,21]. Data were collected from participants living in 42 selected areas consisting of 1,286 contiguous census administrative units (Statistical Areas 1, SA1) across metropolitan and regional cities of seven Australian states/territories. SA1s have an average population size of 400 people. Participants were eligible to participate if they were within the target age bracket, resided at their addresses for at least 6 months prior to the survey, had no physical or intellectual disabilities and provided informed written consent. We used data from the third wave of AusDiab, the only wave in which cognitive function tests were administered and for which relevant environmental exposures were available (ethics approval: Alfred Hospital Ethics Committee, ref. no. 39/11) [21,22,23]. Details about AusDiab data collection procedures are provided elsewhere [20,21,22]. Because this study focused on urban environments, 473 participants who did not reside in urban areas (here defined as towns and cities of 10,000 people or more) were excluded from the analyses, giving an analytical sample of 4141 participants. (The study was conducted according to the guidelines of the Declaration of Helsinki, and approved by the Alfred Hospital Ethics Committee, Melbourne, Australia (ref. no 39/11; 2 March 2011).)

### 2.1. Measures

#### 2.1.1. Environmental Measures (Exposures)

Neighbourhood built and natural environment measures were generated using Geographic Information Systems (GIS) software. Street-network buffers of a 1-km radius were created around geocoded participants’ residential addresses to define neighbourhoods. A 1-km radius was used because it corresponds to a 10–20 min walk, which is a common definition of a neighbourhood [23]. Four built environment, two natural environment and two ambient air pollution measures were computed. These were population density (persons/ha), street intersection density (intersections/km^2^), percentage of commercial land use, non-commercial land use mix (range: 0–1), percentage of parkland, percentage of blue spaces (e.g., lakes, coastlines and rivers) and annual average concentrations of nitrogen dioxide (NO_2_, units: ppb) and fine particulate matter <2.5 µm in aerodynamic diameter (PM_2.5_, units: μg/m^3^). Details on these measures are provided elsewhere [23,24,25] and in the Appendix A.

#### 2.1.2. Cognitive Function Measures (Outcomes)

Memory and processing speed were the cognitive functions examined in this study because they are essential to learning and reasoning, they typically decline with age, but their decline can be slowed down by leading an active lifestyle and reducing cardiometabolic risk factors [26]. Memory was assessed using the California Verbal Learning Test (CVLT) whereby participants recalled 16 common shopping items after a 20-min delay (score range: 0–16) [27]. Processing speed was measured using the Symbol–Digit Modalities test (SDMT), which requires participants to use a reference key to find and report the numbers (1 to 9) corresponding to nine geometric figures as quickly as possible in 90 s (score range: 0–60) [28].

#### 2.1.3. Cardiometabolic Risk Factors (Potential Mediators)

Cardiometabolic risk factors considered as potential mediators of the associations between neighbourhood environment characteristics and cognitive function included: an adiposity indicator (waist circumference expressed in cm); an indicator of elevated blood pressure (mean arterial pressure expressed in mmHg); an indicator of elevated blood glucose (glycated haemoglobin [HbA1c in mmol/mol]); and three indicators of dyslipidaemia (low-density lipoprotein [LDL] cholesterol (mg/dL), high-density lipoprotein [HDL] cholesterol (mg/dL) and triglycerides (mg/dL)). The assessment of cardiometabolic risk factors in AusDiab has been detailed elsewhere [20,21].

#### 2.1.4. Confounders and Covariates

Several variables were included as potential confounders or covariates as appropriate (see Appendix A for details). These were self-reported sex, age, educational attainment, employment status, household income, living arrangements, ethnicity, history of heart problems or stroke, tobacco smoking status, area-level socio-economic status, relevant medications (hypertension, diabetes and lipid-lowering medications) and residential self-selection.

### 2.2. Statistical Analyses

Descriptive statistics and the percentage of missing values were computed for all variables. Because 17% of cases had missing data on at least one variable, multiple imputations by chained equations [29] were used to create ten imputed datasets for the regression analyses. Directed acyclic graphs informed the selection of a minimally sufficient set of confounders for regression models estimating exposure-mediator and mediator-outcome relationships (Appendix A). The potential mediating role of cardiometabolic risk factors in the associations between neighbourhood environmental characteristics and cognitive function was examined using the joint-significance test according to which data support mediation if the exposure-mediator associations and the exposure-adjusted mediator-outcome associations are both statistically significant [15]. Generalised additive mixed models (GAMMs; package ‘mgcv’ version 1.8.22 [30] in R) with random intercepts at the SA1 level were used to estimate these associations to allow for possible curvilinear effects. Analyses were conducted in several steps detailed in the Appendix A.

Briefly, given that our analyses considered potential causal effects among environmental characteristics, we first estimated the confounder-adjusted total effects of each environmental variable on each cardiometabolic risk factor. Here, ‘total effect’ refers to the sum of effects mediated and unmediated by other environmental variables and is estimated by excluding from the regression model those environmental characteristics that are deemed to be in the pathway between the environmental exposure of interest and the response variable (outcome or mediator). We then estimated the ‘direct effects’ of environmental characteristics on the cardiometabolic risk factors (i.e., unmediated by other environmental variables) by including in the regression models all environmental characteristics hypothesised to mediate the effects of the environmental exposure of interest on the response variable. As we hypothesised that adiposity (waist circumference) would be a determinant of other cardiometabolic risk factors [31,32], we also estimated the direct effects of waist circumference on other cardiometabolic risk factors adjusted for all environmental variables. Finally, a set of models estimated the environmental-exposure-adjusted associations of cardiometabolic risk factors with cognitive function. These models also provided estimates of the effects of environmental characteristics on cognitive function not explained by cardiometabolic risk factors (i.e., ‘direct’ effects of the environment on cognition). In the above models, we also examined whether taking medications for a specific cardiometabolic risk factor moderated the associations of environmental characteristics, waist circumferences with the cardiometabolic risk factor and the associations of the cardiometabolic risk factors with cognitive function.

## 3. Results

The average age of the sample was 61 years (SD = 11). The majority of participants were of English-speaking background, female and in paid employment (Table 1). Nearly half of the participants were living with a partner but without children. The sample was heterogeneous in socio-economic status, both in terms of educational attainment and household income. Only 6.3% of participants were taking diabetes medications, while lipid-lowering and antihypertensive medications were considerably more prevalent. There was substantial variability in several neighbourhood environmental characteristics. However, the percentage of blue space and commercial land in residential buffers was low. The average annual concentrations of air pollutants were also relatively low, with NO_2_ at 5.5 ppb and PM_2.5_ at 6.3 μg/m^3^.

The total effects of environmental characteristics on cardiometabolic risk factors are reported in Table 2. Population density and NO_2_ showed mixed total effects on cardiometabolic risk factors. In contrast, intersection density, non-commercial land use mix and PM_2.5_ were prevalently associated with unfavourable outcomes, while indicators of natural environment tended to show positive effects on cardiometabolic health. Taking diabetes medications moderated the total effect of the percentage of blue space on glycated haemoglobin whereby only those on medication showed a negative association (10% increase in blue space; e*^b^* = 0.845; 95% CI: 0.743, 0.962; *p* = 0.011). Significant total but not direct effects on cardiometabolic risk factors were observed for population density in relation to glycated haemoglobin, percentage of commercial land in relation to mean arterial pressure and non-commercial land use mix in relation to HDL cholesterol and triglycerides (Table 2; Figure 2 and Figure 3). The total effects of these three built environment attributes were fully explained by their impact on other environmental attributes depicted in Figure 2 and Figure 3.

HDL cholesterol mediated some of the associations between environmental characteristics and both cognitive functions (Figure 2 and Figure 3), while waist circumference and glycated haemoglobin were identified as mediators of environment-processing speed associations (Figure 3). Environmental attributes that showed positive indirect effects on both cognitive functions through HDL cholesterol were intersection density, population density (via intersection density) and percentage of blue space (via waist circumference) (Figure 2 and Figure 3). Environmental characteristics that exhibited a detrimental indirect effect on cognitive functions through HDL cholesterol were PM_2.5_ and environmental attributes that contributed to higher levels of PM_2.5_ (population density, percentage of commercial land and parkland and non-commercial land use) (Figure 2 and Figure 3). In addition, PM_2.5_ showed a positive direct effect on memory, NO_2_ on processing speed and percentage of parkland on both cognitive functions.

PM_2.5_ and its environmental antecedents had a negative and percentage of blue space a positive, indirect effect on processing speed through waist circumference and glycated haemoglobin (Figure 3). NO_2_ with its environmental antecedents (population density, percentage of parkland, street intersection density and non-commercial land use mix) had a negative indirect effect on processing speed through glycated haemoglobin. Blue space had a positive indirect effect on processing speed through glycated haemoglobin in those on diabetes medications (Figure 3).

## 4. Discussion

This study examined the potential effects of characteristics of the neighbourhood built and natural environment on cognitive function in conjunction with ambient air pollution. In doing so, it identified potential pathways of influence via cardiometabolic health indicators. We discuss the findings starting from the environmental attributes more proximal to cognitive function (ambient air pollution) as depicted in the proposed conceptual model of neighbourhood environmental influences on cognitive health (Figure 1).

### 4.1. Air Pollution

After adjusting for built and natural neighbourhood environmental characteristics, we observed negative indirect effects of PM_2.5_ and NO_2_ on cognitive function via cardiometabolic risk factors. These findings were expected because higher levels of these air pollutants have been previously associated with worse cognitive health outcomes in humans [6,18]. Also, animal models showed that PM_2.5_ and NO_2_ contribute to neurodegenerative processes via cardiovascular and cerebrovascular diseases, and other pathways key to dementia pathogenesis (e.g., Aβ depositions) [18,33]. In this study, the negative effects of PM_2.5_ on cognitive function were channelled through indicators of adiposity (waist circumference), elevated blood glucose (glycated haemoglobin) and hyperlipidaemia (low HDL cholesterol), while those of NO_2_ were mainly explained by elevated blood glucose (glycated haemoglobin). Exposure to air pollution has been previously associated with adiposity [34,35], which, as also suggested by this study, is an established causal factor for elevated blood glucose and dyslipidaemia [36] that, in turn, are harmful to cognitive health [6,37,38,39].

We found negative effects on cognitive function mediated by glycated haemoglobin but independent of waist circumference for NO_2_ but not PM_2.5_. In line with these findings, Honda et al. [40], who estimated long-term annual exposure to air pollutants in older Americans, reported stronger negative effects of NO_2_ than PM_2.5_ on glycated haemoglobin. Similar associations have also been observed in studies of diabetes prevalence [41,42] suggesting that traffic-related air pollution, of which NO_2_ is an indicator, may be more important than particulate matter for glycaemic control [40,41] and may impact cognitive function through these particular cardiometabolic risk factors.

In addition to the negative indirect effects of air pollutants on cognitive function, we observed positive direct effects, which were unexpected and likely due to using coarse measures of land use as proxies of access to destinations promoting cognition-enhancing activities (e.g., places in the neighbourhood for social, physical and intellectual activities). Specifically, commercial destinations that offer opportunities for cognition-enhancing activities (e.g., cultural and entertainment venues, food outlets) can be major sources of pollution generated by food preparation and high volumes of visitors and, hence, traffic [24,25,43]. While we adjusted the effects of air pollutants for proxies of destination accessibility, these proxies (non-commercial land use mix and percentage of commercial land) were likely unable to discriminate destinations for cognition-enhancing activities (and higher levels of pollution; e.g., restaurants) from those that were not (e.g., banks or warehouses). This could have led to us observing the positive direct effects of air pollutants on cognitive function. In fact, positive associations of NO_2_ and PM_2.5_ with walking for transport were observed in an earlier analysis of AusDiab data [23] and negative associations of NO_2_ with mean arterial pressure were found in this study, suggesting that destinations supporting engagement in utilitarian walking and other activities were not accurately captured by the land-use measures used in this study. These findings highlight the need for an accurate and comprehensive characterisation of urban neighbourhood environments, encompassing all key interrelated features, in studies of environmental determinants of cognitive function.

### 4.2. Natural Environment

This study suggests prevalently positive effects of access to parkland on cognitive function, unmediated by cardiometabolic risk factors and mediated by other environmental attributes, albeit some negative indirect effects of parkland via air pollutants and related cardiometabolic risk factors were also observed. Greenspace has been previously found to have positive effects on cognitive function in adults [44], whereas the evidence is less consistent in older adults [45]. The positive effects observed in this study may have been due to various here unmeasured mechanisms, including engagement in physical activity [6,9,14], social activities [46] and attention restoration [47], the latter referring to the restoration of directed attention depleted by attentional tasks for daily living in the context of complex urban environments [47].

Contrary to our expectation that the positive effects of parkland on cognitive function would be in part explained by its mitigating effects on air pollution [48], our measure of greenspace was positively related to both annual average concentrations of PM_2.5_ and NO_2_. The positive association with PM_2.5_ might have been due to natural sources, including wind-blown dust, sea salt and biogenic emissions from parkland [49]. Other parkland-related contributors to neighbourhood-level PM_2.5_ and NO_2_ concentrations could be smoke from prescribed fires [50] and maintenance activities utilising petrol-powered machinery (e.g., grass mowing; leaf blowing) [51]. These potential deleterious effects need to be acknowledged when examining the impact of greenspace on cognitive function and related biomarkers.

While no direct effect of blue space on cognitive function was observed in this study, positive indirect effects through waist circumference, HDL cholesterol and glycated haemoglobin (in those on diabetes medications) were observed. These findings may be due to blue spaces promoting an active lifestyle [52] and physical activity contributing to better cardiometabolic health [53]. Alternatively, as access to blue space promotes outdoor activities [54] that increase exposure to solar ultraviolet radiation [55], the beneficial indirect effects of blue space on cognitive function via waist circumference and glycated haemoglobin, in particular, might have been in part due to higher levels of vitamin D from sun UV exposure [56,57].

### 4.3. Built Environment

We hypothesised that built environment indicators of densification and access to services (e.g., population density, intersection density and land use mix) would show both positive and negative effects on cognitive health given that dense, complex, destination-rich environments provide opportunities for social contacts and other cognition-enhancing activities (e.g., physical activity) but also increase exposures to stressors such as air pollution and noise [17]. In support of this conjecture, we found positive effects as well as air-pollution-mediated negative effects of population and intersection densities on both memory and processing speed via HDL cholesterol. The positive effects of these two built environment attributes may be linked to them facilitating active travel (i.e., walking for transport), as previously observed in this [58] and other cohorts [9,59] and to active travel being associated with healthier blood lipid profiles [60] and better mental and cognitive health [61]. Notably, only after adjusting for air pollutants, we were able to identify the positive and negative effects of population and intersection densities on HDL cholesterol and, thus, cognitive function (NB: the total effects of these attributes on HDL cholesterol were nil). Also, the unfavourable total effects of several built environment attributes on specific cardiometabolic risk factors vanished after adjustment for air pollutants. This showcases the importance of conducting mediation analyses that consider the inter-relationships between activity-promoting features of the urban built environment and its by-products (air pollution) in studies of environmental determinants of cognitive function.

Interestingly, street intersection density and non-commercial land use mix were positively related to cardiometabolic risk factors that did not mediate environment-cognition associations in this study (LDL cholesterol, triglycerides and mean arterial pressure) even after adjustment for air pollutants. Possible contributors to these findings might be access to unhealthy foods (i.e., fast food outlets) [62] and traffic-related and industrial noise [13,63] in neighbourhoods with high levels of non-commercial land use mix (including industrial land) and intersection density [64].

Apart from population density and street intersection density, no other built environmental attribute was directly related to the cardiovascular risk factors associated with cognitive function in this study. However, the percentage of commercial land and land use mix were related to air pollutants which, in turn, showed negative effects on cognitive function mediated by cardiometabolic risk factors and positive effects not mediated by cardiometabolic risk factors. As explained earlier, the latter effects were likely due to air pollutants being indicators of human activities and, hence, opportunities to engage in cognition-enhancing pursuits (socialising, entertainment, intellectual activities) rather than them being beneficial to cognitive health. These findings not only highlight the need for simultaneously examining the effects of key interrelated features of urban environments on cognitive function but also the need for a more fine-grained characterisation of urban environments in terms of destinations that provide attractive opportunities for cognition-enhancing activities (e.g., the density of food outlets, places for socialising, quality of green spaces). This information is essential for the cognitive health impact assessment of neighbourhood environmental features that, in turn, can guide city planning policies.

### 4.4. Strength and Limitations

A strength of this study is the utilisation of data from a national cohort capturing diverse urban environments in Australia. We examined the joint linear and/or curvilinear effects of the neighbourhood built and natural environment and air pollution on cognitive function, and the potential cardiometabolic mechanisms underpinning them. By doing so, we accounted for neighbourhood self-selection to partially address reverse causality. We examined total, direct and indirect cross-sectional effects of urban environmental features on cognitive function to disentangle their potentially beneficial and harmful impacts. Study limitations include the cross-sectional nature of the available data and the employment of coarse measures of destination accessibility relevant to cognition-enhancing activities. The latter limitation has made it difficult to disentangle the positive (access to destinations supporting healthy behaviours) and negative effects (pollution and access to destinations promoting unhealthy behaviours) of urban densification on cognitive function and related cardiometabolic risk factors. It would have also been desirable to have data on other key covariates, such as estimates of indoor pollution levels in the home. Future research would need to address these limitations by conducting longitudinal studies able to capture changes in exposures and cognitive function [65] and more accurately characterise urban environments that may influence key biological and behavioural risk factors of cognitive decline.

## 5. Conclusions

In line with a proposed ecological model of neighbourhood environmental influences on cognitive health, this study has found features of urban environments to be directly or indirectly related, via cardiometabolic risk factors, to cognitive function in middle-aged and older Australian adults. Dense, interconnected neighbourhoods may contribute to better cardiometabolic outcomes (e.g., higher HDL cholesterol) and, consequently, better cognitive health by promoting active transportation. While such environments usually provide better opportunities for social contacts and other cognition-stimulating activities, they are often sources of air pollution arising from human activities and vehicular traffic that harm cardiometabolic health and, hence, cognitive function. Our findings also provide support for the positive effects of green and blue spaces on cognitive function via cardiometabolic risk factors and other mechanisms that were not examined in this study (e.g., attention restoration, social contacts or physical activity). Cognition-friendly urban environments appear to be typified by compact, interconnected neighbourhoods with good access to green and blue spaces and low average annual levels of PM_2.5_ and NO_2_. Longitudinal studies with a more accurate characterisation of the built environment, the quality of natural spaces and individual activity locations (within and outside the neighbourhood) are needed to better understand how to create cities that can help preserve cognitive function in ageing populations.

## Figures and Tables

**Figure 1 toxics-10-00023-f001:**
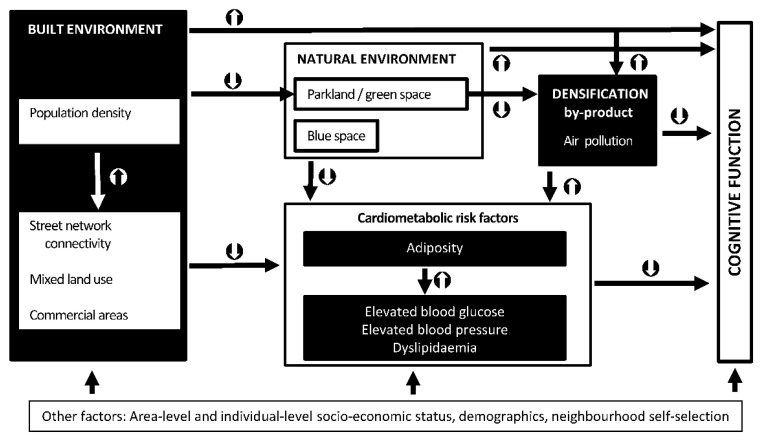
A simplified ecological model of neighbourhood environmental influences on cognitive function. 
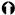
 indicate positive associations; 
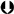
 indicate negative associations.

**Figure 2 toxics-10-00023-f002:**
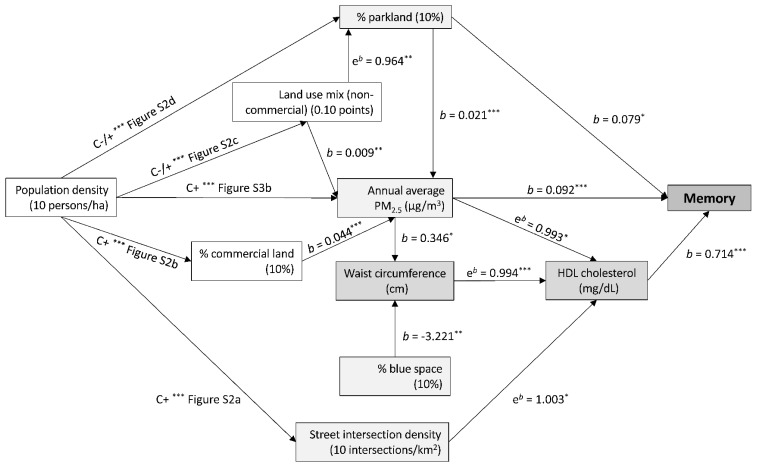
Effects of neighbourhood environmental characteristics on memory and the mediation roles of cardiovascular risk factors. Arrows linking variables indicate significant associations; *b*, regression coefficient; e*^b^*, exponentiated regression coefficient; ha, hectare; C+, −, +/− = curvilinear positive, negative, non-monotonic association; Figure Sx; the supplementary figure of a curvilinear association. * *p* < 0.05; ** *p* < 0.01; *** *p* < 0.001. Cardiometabolic risk factors are represented by darker grey rectangles; environmental characteristics directly associated with memory or cardiometabolic risk factors are represented by light grey rectangles; environmental attributes indirectly associated with cardiometabolic risk factors through other environmental characteristics are represented by white rectangles. All significant and non-significant associations (regression coefficients and 95% CIs) are presented in the Appendix A. Appendix A also provides the F-ratio of the smooth terms for significant curvilinear associations.

**Figure 3 toxics-10-00023-f003:**
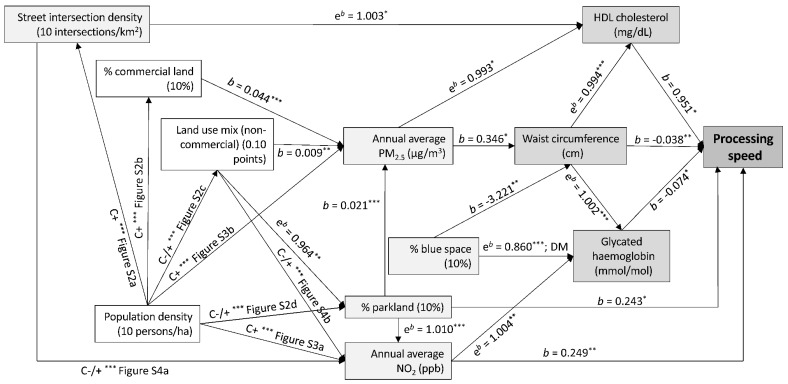
Effects of neighbourhood environmental characteristics on processing speed and the mediation roles of cardiovascular risk factors. Arrows linking variables indicate significant associations; *b*, regression coefficient; e*^b^*, exponentiated regression coefficient; ha, hectare; C+, −, +/− = curvilinear positive, negative, non-monotonic association; Figure Sx; supplementary figure of a curvilinear association. * *p* < 0.05; ** *p* < 0.01; *** *p* < 0.001. Cardiometabolic risk factors are represented by darker grey rectangles; environmental characteristics directly associated with memory or cardiometabolic risk factors are represented by light grey rectangles; environmental attributes indirectly associated with cardiometabolic risk factors through other environmental characteristics are represented by white rectangles. All significant and non-significant associations (regression coefficients and 95% CIs) are presented in the Appendix A. Appendix A also provides the F-ratio of the smooth terms for significant curvilinear associations.

**Table 1 toxics-10-00023-t001:** Sample characteristics (N = 4141).

Characteristics	Statistics	Characteristics	Statistics
**Socio-demographic characteristics**			
Age (years), M ± SD	61.1 ± 11.4	Sex, female, %	55.2
Educational attainment, %		Employment status, %	
Up to secondary	32.7	Not employed	30.4
Trade, associate diploma	43.6	Paid employment	52.2
Bachelor degree, postgraduate	23.1	Volunteering	15.1
Missing data	0.6	Missing data	2.3
Living arrangements, %		Household income (annual), %	
Couple without children	48.2	Up to $49,999	32.9
Couple with children	26.8	$50,000–$99,999	26.8
Other	22.4	$100,000 and over	28.9
Missing data	2.4	Missing data	11.5
Area-level IRSAD, M ± SD	6.4 ± 2.7	English-speaking background, %	89.9
Residential self-selection—access to destinations, M ± SD	3.0 ± 1.4	Residential self-selection—recreational facilities, M ± SD	3.1 ± 1.5
Missing data, %	7.8	Missing data, %	7.8
**Cardiometabolic risk factors and other health-related variables**			
Heart problems/stroke history, %	8.7	Tobacco-smoking status, %	
Missing data, %	1.0	Current smoker	7.0
LDL cholesterol, mg/dL, M ± SD	3.0 ± 0.9	Previous smoker	35.9
Missing data, %	1.4	Non-smoker	54.5
HDL cholesterol, mg/dL, M ± SD	1.5 ± 0.4	Missing data	2.6
Missing data, %	0.3	Waist circumference (cm), M ± SD	94.6 ± 14.2
Triglycerides, mg/dL, M ± SD	1.3 ± 0.9	Missing data, %	0.2
Missing data, %	0.3		
Glycated haemoglobin (HbA1C), mmol/mol, M ± SD	39.9 ± 6.3	Mean arterial pressure, mmHg, M ± SD	92.0 ± 12.3
Missing data, %	0.5	Missing data, %	0.2
Diabetes medication, %	6.3	Anti-hypertensive medication, %	32.0
Missing data, %	1.8	Missing data, %	1.8
Lipid-lowering medication, %	24.5		
Missing data, %	1.8		
**Cognitive function, M ± SD**			
Memory, CVLT score	6.5 ± 2.4	Processing speed, SDMT score	49.7 ± 11.6
Missing data, %	2.3	Missing data, %	2.0
**Neighbourhood environmental characteristics (1 km-radius street-network buffers), M ± SD**			
Population density, persons/ha	17.4 ± 10.0	Street intersection density, intersections/km^2^	62.2 ± 32.2
Percentage of commercial land use in residential buffer	2.5 ± 6.1	Non-commercial land use mix, entropy score (0 to 1)	0.14 ± 0.13
Percentage of parkland in residential buffer	11.6 ± 12.5	Percentage of blue space (waterbody) in residential buffer	0.24 ± 1.98
NO_2_, ppb	5.5 ± 2.1	PM_2.5_, μg/m^3^	6.3 ± 1.7

Notes. M, mean; SD, standard deviation; IRSAD, Index of Relative Socioeconomic Advantage and Disadvantage; LDL, low-density lipoprotein; HDL, high-density lipoprotein; CVLT, California Verbal Learning Test; SDMT, Symbol–Digit Modalities Test; NO_2_, nitrogen dioxide; PM_2.5_, particulate matter <2.5 µm.

**Table 2 toxics-10-00023-t002:** Total effects of neighbourhood environmental characteristics on cardiometabolic risk factors.

Environmental Characteristic (Units)	Waist Circumference (cm)	HDL Cholesterol (mg/dL)	LDL Cholesterol (mg/dL)	Triglycerides (mg/dL)	Glycated Haemoglobin (mmol/mol)	Mean Arterial Pressure (mmHg)
	*b* (95% CI)	*e^b^* (95% CI)	*b* (95% CI)	*e^b^* (95% CI)	*e^b^* (95% CI)	*b* (95% CI)
Population density	−0.163	0.998	−0.011	0.992	**1.006**	−**0.659**
(10 persons/ha)	(−0.646, 0.320)	(0.989, 1.008)	(−0.037, 0.014)	(0.972, 1.012)	**(1.002, 1.010)**	**(−1.083, −0.234)**
Street intersection density	0.145	1.001	**0.010**	**1.012**	1.000	**0.706**
(10 intersections/km^2^)	(−0.035, 0.326)	(0.998, 1.005)	**(0.001, 0.019)**	**(1.004, 1.019)**	(0.999, 1.002)	**(0.553, 0.859)**
Percentage of commercial land	0.270	0.986	−0.010	1.026	0.998	**0.761**
(10%)	(−0.504, 1.044)	(0.972, 1.001)	(−0.051, 0.031)	(0.993, 1.060)	(0.991, 1.004)	**(0.086, 1.435)**
Non-commercial land use mix	0.377	**0.990**	0.014	**1.015**	1.002	**0.761**
(0.10 score)	(−0.004, 0.770)	**(0.983, 0.997)**	(−0.006, 0.034)	**(1.000, 1.031)**	(0.999, 1.006)	**(0.424, 1.098)**
Percentage of parkland	−0.314	0.996	**−0.021**	1.003	1.001	−0.303
(10%)	(−0.714, 0.087)	(0.989, 1.004)	**(−0.041, −0.001)**	(0.987, 1.019)	(0.997, 1.004)	(−0.660, 0.053)
Percentage of blue space	**−3.237**	**1.041**	−0.076	0.972	0.989 *	−1.566
(10%)	**(−5.461, −1.013)**	**(1.003, 1.088)**	(−0.197, 0.045)	(0.882, 1.070)	(0.971, 1.006)	(−3.477, 0.346)
NO_2_	0.066	0.999	−0.002	1.000	**1.003**	**−0.381**
(ppb)	(−0.242, 0.373)	(0.993, 1.006)	(−0.018, 0.015)	(0.988, 1.013)	**(1.000, 1.006)**	**(−0.647, −0.115)**
PM_2.5_	**0.372**	**0.994**	**0.026**	1.008	1.001	**Curvilinear** (see Appendix A)
(μg/m^3^)	**(0.064, 0.681)**	**(0.990, 0.999)**	**(0.006, 0.047)**	(0.996, 1.021)	(0.998, 1.006)	

Notes. *b*, regression coefficient; *e^b^*, exponentiated regression coefficient; CI, confidence intervals; * effect moderated by diabetes medication. Effects in bold are statistically significant at a probability level of 0.05. Details on regression models, including confounders, are in the Appendix A.

## Data Availability

Data that support the findings of this study are available on request under a license agreement. Written applications can be made to the AusDiab Steering Committee (Dianna.Magliano@baker.edu.au).

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
