# Peer review of "Urban Neighbourhood Environments, Cardiometabolic Health and Cognitive Function: A National Cross-Sectional Study of Middle-Aged and Older Adults in Australia"

_toxics, 2022, doi:10.3390/toxics10010023_

Round 1
Reviewer 1 Report
This is a very interesting and well written paper. Since my expertise is related to the built environment, I cannot thoroughly assess the statistical analysis of the paper, but it seemed fine to me. The authors themselves point out the coarse characteristics they use to differentiate between different built environments, and I can only agree that it is very difficult to identify measurable factors that give a reliable qualification of good versus bad neighborhoods. For me the discussion of these factors, and the mediation with air pollution, is a positive point in this paper.
What is missing on an overall level, is the identification of potential social factors that also play a role in the relation between built environment and cognitive health. It might for example be important to identify where people's children live, and how the distance to the children might play a role. The study however explains very well its limitations, so I don't have any trouble approving it for publication.
Reviewer 2 Report
The manuscript provides a detailed and remarkably clear analysis of the impact of multiple environmental factors on cognitive function. The authors are to be complemented on their thoroughness, both in the analysis and in the presentation of the results. The use of mediation analysis to examine the interactions among multiple neighborhood characteristics is very much needed to inform our understanding of neighborhood characteristics on all aspects of health.
I have a few minor suggestions on the manuscript and supplemental materials:
Results
Table 2. The authors use an asterisk (*) to identify effects moderated by diabetes medication. A comma is needed in line 230 between "confidence intervals" and the asterisk. It would be helpful to the reader if the font of the asterisk was increased both in line 230 and in the table where it appears.
End matter:
Lines 434-438. "Data Availability Statement". It appears the authors have retained the instructions from the journal. This section either needs to be removed or replaced with language relevant to the paper.
Supplemental Materials
S2. Detailed description of analytical steps: In the text and in Table S1 does "T" refer to "Total Effects". Similarly, does "D" refer to "direct effects" or "density". Please, clarify in both the text and with a comment in the footnotes to the table.
Author Response
Apologies. While I was uploading our response to this Reviewer's comments (originally labeled Reviewer 1), comments from another Reviewer 1 appeared online and I accidentally submitted the responses to the comments from the Reviewer (now Reviewer 2) under Reviewer 1. Therefore, here I have attached the comments to both reviewers (Reviewer 1 and Reviewer 2). Please see attachment.

Reviewer 3 Report
This is a very interesting paper which describes a national cross-sectional study on the links between environmental parameters and cognitive functions of 4,141 middle-aged and older Adults in Australia. This is a carefully done work and the findings are of considerable interest enough to merit publication. Before acceptance, please consider following points.
- As shown in Table 1, average annual concentration of PM2.5 was 6.3±1.7 µg/m3 which is much lower than WHO guideline level of 10 µg/m3 and seems to be background level. Previous studied have shown the negative effects on the cardiometabolic risk factors, including adiposity [34, 35]. However, the studies in the references [34, 35] included or used data of highly polluted regions such as China and south Korea. So, my simple question is whether such lower levels of PM2.5 can actually contribute to the cardio-metabolism of residents in Australia.
- I have also concern on indoor levels of PM2.5 in the studied residences. When the outdoor level is low, indoor PM2.5 can be a significant source of exposure. Please mention about the relationship between outdoor and indoor levels of PM2.5 in the studied area.
- I think the dietary pattern (food habit) is a significant factor for the cardiometabolic risk. How did authors consider on this point in the study design.
Reviewer 4 Report
I find the study as described by the manuscript highly relevant to public health issue using a comprehensive data set and detailed statistical analysis to disentangle the different variables that can effect the cognitive function of subjects. The methods used in the study are sound and rigorous in data analysis. As the study only focussed on built-in environment of urban areas and hence excluded non-urban areas from the analysis in order to build a generalised additive mixed models (GAMM) for linear and nonlinear models with built-in urban environment variables. The authors should also consider an alternative question: can non-urban areas be used as the control to determine the effect of built environment on cognitive function ? . This could be simpler as it avoid the need to account for some compouding variables and provide some insightful results.
I recommend the manuscript to be accepted for publication with some minor revisions
Specific comments
(1) Line 160: 'the SA1 level". Please specify full name of SA1 for people who is not familiar with Australia census terminology, and give the sense of spatial scale in the data
(2) Line 312: "smoke from prescribed fires" ?. I thought that prescribed fires are prohibited in parklands in Australia
(3) Line 313: "grass mowing; leaf blowing". These activities caused only short term and at neighbourhood level and wont effect annual average NO2 or PM2.5 concentration
